# Effectiveness of targeted antenatal family planning information provision on early postpartum family planning uptake in Kisumu County: Protocol for a simple randomized control trial (I DECIDE Study)

**Morris Senghor Shisanya**[1¤a]*, **Collins Ouma**[2⊙], **Mary Kipmerewo**[3⊙¤b]

**1** Department of Community Health Nursing, School of Nursing, Kibabii University, Bungoma, Kenya,
**2** Department of Biomedical Sciences and Technology, Maseno University, Maseno, Kenya, **3** Department of Reproductive Health, Midwifery and Child health, School of Nursing, Midwifery and Paramedical Sciences (SONMAPS), Masinde Muliro University of Science and Technology (MMUST), Kakamega, Kenya

⊙ These authors contributed equally to this work.
¤a Current address: School of Nursing, Kibabii University, Bungoma, Kenya
¤b Current address: School of Nursing, Midwifery and Paramedical Sciences (SONMAPS), Masinde Muliro University of Science and Technology (MMUST), Kakamega, Kenya
* mshisanya@kibu.ac.ke

**Data Availability Statement:** Type of Access: Controlled Access, Type of analysis: Quantitative

## Abstract

Overlooking the contraceptive needs of postpartum women constitutes missed opportunities in health system. Inter-birth interval of at least three years can prevent poor maternal, perinatal and neonatal outcomes and afford women socio-economic benefits of family planning (FP). The unmet need for FP in the postpartum period remains unacceptably high and far exceeds the FP unmet need of other women. The Kenya Demographic and Health Survey (KDHS) estimate the unmet need for postpartum FP to be 74%. Maternal and Child Health (MCH) continuum provides a great opportunity for postpartum FP (PPFP) interventions integration especially antenatal targeted FP information giving and gauging of fertility intentions. However, there is no protocol for structured, targeted antenatal FP information giving and behavioural contracting to influence postpartum fertility intentions of mothers before delivery. Knowledge gap regarding fertility intentions and best antenatal strategies for postpartum FP still exists. The available evidence differs across settings and demography. Equally, there has been inadequate exploration of operationally-feasible ways to integrate FP counselling into existing ANC services with limited number of methodologically rigorous trials. The current protocol will therefore examine the effectiveness of targeted antenatal family planning information provision on early postpartum FP uptake using a randomized control trial in Kisumu County, Kenya. The protocol will assess socio-cultural beliefs towards PPFP and perceived individual control of PPFP choice, analyze knowledge and intention for PPFP, and finally compare and examine the determinants of PPFP uptake between study groups. Through simple sampling, a group of 246 antenatal mothers will be randomly assigned to control, community and facility intervention groups as per eligibility criteria in the study facilities. After at least 3 months of intervention and postpartum follow-up,

analysis Process of Requesting for Data: Request to be done by email of the principal investigator The decision is by the Research team lead by the Principal investigator Criteria for reviewing the request: Qualification of the person requesting, description of the purpose of the request, willingness to engage the Principal investigator in the development of analysis plan, and monographs or manuscripts.

**Funding:** The author(s) received no specific funding for this work.

**Competing interests:** The authors have declared that no competing interests exist.

**Abbreviations: ANC**, Antenatal Care; **CHV**, Community Health Volunteers; **CRF**, Case Report Form; **CU**, Community Unit; **FP**, Family Planning; **IERC**, Institutional Ethics and Research Committee; **KDHS**, Kenya Demographic Health Survey; **KNBS**, Kenya National Bureau of Statistics; **MMUST**, Masinde Muliro University of Science and Technology; **NACOSTI**, National Commission for Science, Technology and Innovation; **PPFP**, Postpartum Family Planning; **RCT**, Randomized Control Trial; **SPSS**, Statistical Package for Social Sciences.

clinical superiority will be used to gauge which intervention was effective and the model superiority. Questionnaire and Case Report Forms will be the main source of data. The participant will form the unit of analysis which will be by intention to treat. Bivariate analysis will be applied as the selection criteria for inclusion of predictors of intention and uptake in the final logistic regression model. Odds Ratios and 95% confidence interval (CI) will be used to demonstrate significance and the strength of association between selected variables. Dissemination will be through conference presentations and peer reviewed journals. The trial has been registered with the Pan African Clinical Trials Registry PACTR202109586388973 on the 28th September 2021.

## Introduction

Inter-birth interval of at least three years by use of effective postpartum contraceptive methods in less developed countries could prevent poor maternal, perinatal, and neonatal health outcomes, including stillbirth, prematurity, low birth weight, neonatal and maternal mortality [1, 2].

Ignoring the contraceptive needs of postpartum women constitute missed opportunities in health service delivery to avert unplanned pregnancies and afford every woman and her family the health, social, and economic benefits of family planning (FP) [3]. The unmet need for FP in the postpartum period remains unacceptably high and far exceeds the FP unmet need of the rest of the women of reproductive age [4, 5].

Globally, approximately 44% of pregnancies are unintended. The unintended pregnancy rates have declined by 30% in developed regions, from 64 per 1000 Women of Reproductive Age (WRA) in 1990s to 45 by 2018. In developing regions, the unintended pregnancy rate reduced by 16% in the same period to 65%. Decline in the unintended pregnancy rate in both regions coincided with increased uptake of family planning [6]. Inter-birth intervals in 50% or more of pregnancies in low-income countries are too short at less than 23 months [4]. The proportion of postpartum women who no longer want children or want to postpone another child for at least two years but are not using a contraceptive method is still high. The unmet need soon after birth even reaches 75% for the West and Central Africa region [7].

Despite observed advances in access to Reproductive, Maternal, Neonatal and Child Health (RMNCH) services in Kenya in recent decades, the situation of postpartum FP (PPFP) is not different from other low-income countries. In a multi-country study, only 25% of women in Kenya had adopted PPFP by six months, and 35% at one year [8]. This points to feeble progress when it comes to effective postpartum contraceptive service integration [9, 10]. Likewise, the study region, has some of the adverse national indicators of poor uptake of PPFP including; short inter-birth intervals and high fertility rates for women aged 25–40 years and low postpartum care provision including PPFP (63%), as compared to other Counties that were studied (80%)[10, 11].

## Problem statement

Routine antenatal services offer frequent points of contact for providers and pregnant women and Achyut *et al.*, (2016) recommended integration of PPFP information given during ANC period to accelerate early uptake and reduce the high unmet PPFP need in Kenya [4, 12]. These contacts are vital opportunities for information giving, counselling and behavioural contracting to address the contraceptive needs of postpartum women [13]. However,

the operationalization of integration FP interventions during this period remains overlooked at different levels of planning in the Kenyan health care system [14–17]. There is no structured FP information giving, counselling and behavioural contracting to ensure earliest uptake of PPFP. In addition, there is still paucity of evidence that this structured integration can work. Likewise, evidence is often weak or incomplete particularly regarding studies that explore the desires, intentions, and priorities of women or couples related to PPFP, which may differ across settings and demography. Correspondingly, there has been inadequate exploration of operationally-feasible ways to integrate PPFP information giving and counselling into existing ANC services. As such, the proposed protocol will examine the effectiveness of targeted antenatal family planning information provision on early postpartum family planning uptake.

## Justification

This study proposes to bridge the gap of limited exploration of operationally-feasible ways of integrating PPFP in ANC services using a methodologically rigorous interventional study.

The study will examine the effectiveness of integrating FP information provision on PPFP uptake by use of a rigorous but simple randomized control trial. The proposed methods were designed based on feasibility of adoption in ANC practice if proven to be clinically superior. Kisumu County was considered a suitable site for the study as it has some of the adverse outcomes of low early PPFP uptake. Conversely, the County has a functional community health services and primary health facilities linkage which will enhance the feasibility of the study.

## Study aim

Broadly, the study will examine the effectiveness of targeted antenatal family planning information provision on early postpartum family planning uptake in Kisumu County, Kenya. The study will specifically: evaluate effect of socio-cultural beliefs on Postpartum Family Planning uptake among postpartum mothers; assess perceived individual control of Postpartum Family Planning choice among postpartum mothers; analyse fertility intentions for postpartum mothers after the intervention; compare Postpartum Family Planning uptake between control and intervention groups of postpartum and examine the determinants of Postpartum Family Planning uptake between control and intervention groups of postpartum mothers in Kisumu County, Kenya.

## Methods/Design

### Study design

This will be a prospective interventional study, a Randomised Control Trial (RCT) conducted in Kisumu East Sub-County within Kisumu County. The randomly sampled facilities are Migosi and Gita Health Centres for intervention in urban and rural areas, respectively. The Community Units (CU) for community-based intervention are Kuoyo CU and Nyalunya CU in an urban and rural areas, respectively. The control facility and CUs are Kowino and Chiga Health Centres and their link CUs in urban and rural areas, respectively.

The study will have three arms: facility intervention arm, community intervention arm and a control arm. The study will have three interacting phases (Figs 1 and 2): The pre-intervention phase, intervention phase and post-intervention phase. The proposed methods in each phase are not complex thus the overall design can be classified as simple interventional design.

The pre-intervention phase will be for establishing sampling frame, intervention package and research tools formulation.

| | STUDY PERIOD | | | | | |
|---|---|---|---|---|---|---|
| | Enrolment | Allocation | Post-allocation | | Close-out | |
| **TIME POINT\*\*** | - $t_1$ | $t_0$ | $t_1$ | $t_2$ | $t_{x1}$ | $t_{x2}$ |
| | 20/02/2022 | 20/03/2022) | 20/03/2022 | 20/04/2022 | 20/04/2022 | 20/08/2022 |
| **ENROLMENT:** | | | | | | |
| Eligibility screen (32 weeks Pregnant | X | | | | | |
| Informed consent | X | | | | | |
| Allocation | | X | | | | |
| **INTERVENTIONS:** | | | | | | |
| Targeted FP Counselling-Facility Based | | | X | X | | |
| Targeted FP Counselling-Community Based | | | X | X | | |
| Control Group – Routine Ante-Natal Care | | | X | X | | |
| **ASSESSMENTS:** | | | | | | |
| *Demographic aspects, Pregnancy related aspects, Co-morbidities – Through Case Report Forms (CRF)* | X | X | | | | |
| *Future fertility Intention – setting date for postnatal FP visit during ANC period – Exit Interviews* | | | X | X | | |
| Knowledge of family planning methods – Exit Interview | | | X | X | | |
| Perceived control of fertility decision | | | X | X | | |
| Attitude toward PPFP | | | X | X | | |
| *Uptake of Postpartum FP – Final Questionnaire* | | | | | X | X |

**Fig 1. SPIRIT schedule for enrolment, allocation, post-allocation and close-out.**

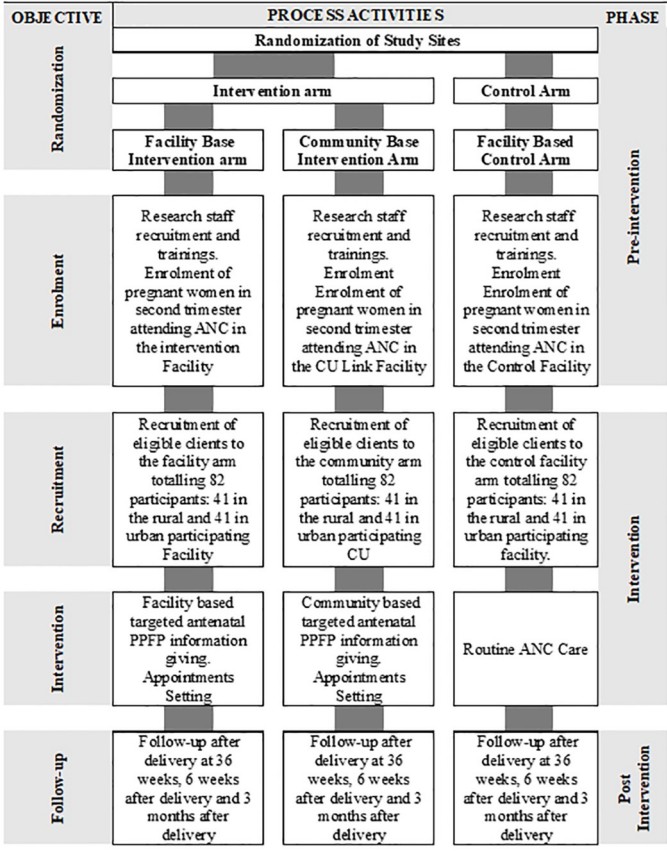

**Fig 2. Phases of the clinical trial.**

The aim of the intervention phase is to determine the effect of the targeted antenatal PPFP information package on the uptake of contraceptive methods during the postpartum period, in comparison with the standard of care. The intervention phase will also integrate qualitative research that is aimed at identifying operational barriers and enablers of the intervention outcomes.

Randomization to control and intervention arms is at individual level within randomly selected control, community and the primary level health centres. Participants allocated to the experimental group will receive the targeted antenatal PPFP information package and those allocated to the control group will receive usual antenatal care.

The intervention will be provision of antenatal information on PPFP using a standardized PPFP counselling tool (S1 Video) and postpartum appointment setting. The control group will be under the routine standard antenatal care. There will be antenatal PPFP information provision training for the service providers at the facility and Community Health Volunteers (CHV) for the community arm to standardize the intervention.

## Study population

This study will be among pregnant mothers in their second trimester, irrespective of age, attending ANC clinic in the intervention facilities or who are within the respective CUs followed up to 3 months postpartum. Excluded will be those who meet the eligibility criteria but are in another similar study, have latex sensitivity, not anticipating male partner in the next 12 months or the only male partner has had a vasectomy. The study will be conducted in 2 primary health centres and 1 community in each of the two sub-counties. The health centres eligibility depended on if: they offer the continuum of ANC, delivery, and Postnatal Care (PNC); they provide a selection of at least three modern contraceptive methods that are rated 2 or 1 on the Medical Eligibility Criteria (MEC) for postpartum contraceptives and there were no stockouts of contraceptives during the preceding six months.

## Sampling method

Multistage sampling will be applied involving purposive sampling for the sub counties, purposive sampling for the intervention and control facilities and community and simple random sampling for the subjects. One sub-county will be purposively sampled by the researchers; one with a rural and an urban set up and should be within easy reach by the team.

The facilities will be matched based on (1) the average number of deliveries per month and (2) the operational level. The intervention CU would be linked to a facility meeting the basic criteria for intervention facility. Each client meeting the criteria will then be randomly assigned to the study i.e. to the intervention and to the control arms using simple random sampling by picking folded paper labelled "yes" or "no". There will be no blinding as the intervention facilities and controls are already known.

## Sample size determination

The sample size is estimated based on the following assumptions: among women at three months postpartum, KDHS data report 27% of use of any method (modern or traditional) in Kenya while the Contraceptive Prevalence Rate (CPR) in the general population is 53% [18]. These figures allow the assumption of a desired 26% difference between control and the intervention groups (26% increase in adopting a modern contraceptive by three months postpartum). Therefore, the sample size was calculated pairwise for two separate RCT for community—control arm and the facility—control arm [19–22]. Rosner, 2015 proposed the sample size determination formula for difference in proportions with consideration of type I and II

errors and power [22, 23], to be:

$N1 = \{z_{(1-\alpha/2)} * \sqrt{(\bar{p} * \bar{q} * (1 + 1/k))} + z_1 - \beta * \sqrt{(p1 * q1 + (p2 * q2)/k)}\}^2 / \Delta^2$. Where q1 = 1-p1, q2 = 1-p2, $\bar{p} = (p1 + kp2)/(1 + K)$, p1, p2 = proportion (incidence) of groups 1 (27%) and #2 (53%), $\Delta$ = |p2-p1| = absolute difference between two proportions (0.26), n1 = sample size for group #1, n2 = sample size for group #2, $\alpha$ = probability of type I error (is set at 0.05), $\beta$ = probability of type II error (is set at 0.1 i.e. 90% power), z = critical Z value for a given $\alpha$ or $\beta$(1.96) and K = ratio of sample size for group #2 to group #1(1). Thus for practical equal sample distribution, the actual sample size shall be 246. As such, each facility shall have 41 clients.

## Intervention procedure

The study will have 3 phases; pre-intervention (-t for enrolment and $t_0$ for allocation), intervention ($t_1$-$t_2$ for post-allocation) and post-intervention ($t_{x1}$-$t_{x2}$ for close out). SPIRIT schedule of the study and a flow diagram of the study are shown in Figs 1 and 2 respectively. The estimated starting date of the study will be 20[th] February, 2022 and the estimated ending date will be in August, 2022.

The intervention will be; provision of antenatal information on PPFP using a standardized PPFP counselling tool and postpartum appointment setting. The intervention will be administered once. This will be within the second and third trimesters of pregnancy. The service providers at the facility and CHVs for the community arm will be trained on antenatal PPFP information provision prior to recruitment. To standardize the training to all, pre and post-test will be administered and the data collectors will be expected to get at least 60% in post-test for them to be brought on board as study implementers. Their post-test score will be included in the analysis to rule it out as a confounder.

The study nurses in all the arms will do the recruitment of clients through history-taking and carrying out physical assessment. The nurse will then fill the case report form. The control group will be under the routine antenatal care after recruitment. The study nurse or CHV will use a standardized tool of Medical Eligibility Criteria that has been embedded in the teaching tool to deliver the intervention. Each session will be standardized to at least 20 minutes for it to be considered adequate information provision.

All the clients will then be undertaken through the exit interview and informed about the last follow-up interview which is between 14 to 16 weeks postpartum.

Client exit interviews entries to the data base and the dashboard will be reviewed by the data management team to give feedback to the study steering committee for appropriate corrective actions to improve adherence to intervention protocols.

There is no concomitant care or intervention that is recommended or prohibited during the trial that has not been contemplated in the exclusion criteria.

Client will be discontinued if they lose their only sexual partner or the only sexual partner undergoes vasectomy in the course of pregnancy, and if the client develops postpartum psychosis, or is hospitalized for more than 14 weeks postpartum.

## Data collection procedure

Five tools will be used for data collection, namely; client exit interview guide, case report form, appointment card, site appraisal form and questionnaire. All the tools will be used to collect quantitative data except site appraisal form or some questions in the questionnaire that need brief explanation. The theory of planned behaviour (Fig 3) was applied to design quantitative process and outcome indicators and thus the same tools will be applied [24]. Client exit interview guide and site appraisal form will be developed based on the procedures set out in the

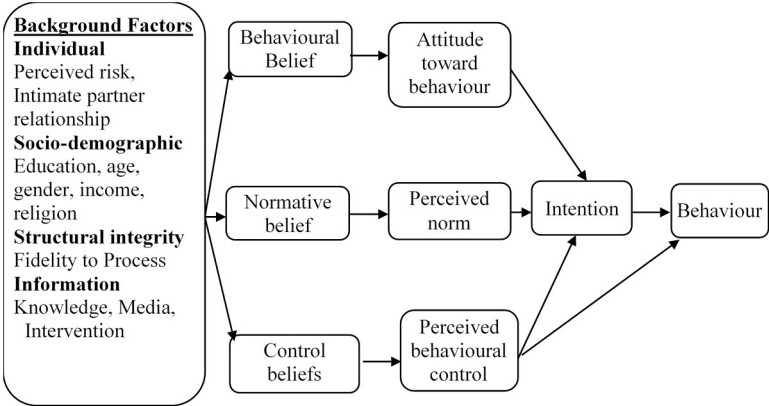

**Fig 3. Theory of planned behaviour schema.**

counselling guide. Appointment card will be the source of information on client details, proposed date for PPFP initiation and vital PPFP information summary.

Each health centre and community unit will have a trained research assistant. The assistant is to ensure adherence to the study manual and standard operating procedures for data management. The primary outcomes will be PPFP knowledge, the intent for use of PPFP (behavioural contracting) and the actual uptake of modern contraceptive methods at three months postpartum. The outcome will largely be assessed based on the CRFs, appointment card and questionnaire. Actual uptake of modern contraceptive method will be established three months postpartum between the 12th to 14th weeks after birth. The CRFs will be filled on recruitment by the trained ANC service provider. The appointment card will be filled by the health worker after the client has accepted to set postnatal follow up date for PPFP. The questionnaire will be filled at 14 weeks postpartum during the scheduled MCH visit by a trained enumerator.

A process evaluation will be undertaken with the objectives of understanding the barriers and enablers related to the delivery of PPFP. This will be evaluated based on client exit interviews and site appraisal forms. The client exit interview guide and site visit appraisal form will be used to assess the process quality indicators which will include: waiting time, time/trimester of start of FP counselling, time after FP counselling, group/ individual session, availability of teaching aid on the table during counselling, gauging FP information level, application of GATHER (Greet, Ask, Tell, Help, Explain and Return/Refer) Model and BRAIDED (Benefits, Risks, Alternatives, Inquiries, Decision, Explanation, Documentation) Model. The models will capture satisfaction with information given, responses to questions, respectful care, confidentiality and privacy. Client exit interviews will be done by trained enumerator immediately after the intervention has been administered to reduce recall bias by the client. Site appraisal form will be filled on site visits by the research team and the enumerator will fill other details to include workload for ANC, number of staff available to offer ANC services, availability of FP counselling bag and flipchart.

Internal consistency will be ensured by piloting the tools and refining them to ensure they capture the essence of what they were meant to collect and Cronbach's alpha of 0.7 will be acceptable.

## Data management

The data will be password-protected: only authorized users will be allowed access to the data. In addition, the collected information will be stored safely; hardcopies will be stored in lockable cabinets and soft copies will be secured by password. Data transmission will be encrypted

to ensure data integrity and confidentiality of participants. Data entry will be done in SPSS version 26.

Data management team will comprise the PI, site data clerks, the study nurses and Community Health Workers. The main roles of team will include; monitoring acceptance rate (proportion of those who accepted to participate versus those sampled), monitoring retention (percentage of participants proceeding from recruitment, treatment to follow-up including missing outcome data), monthly forecast of recruitment for the trial period remaining, monitoring loss to follow-up as a proportion of the ones not yet reached by 16 weeks postpartum follow-up to those who were recruited, overseeing data management metrics such as the rate of electronic data capture, return dates and rate of returns, number of completed follow-up, withdrawal rate, monitoring intervention fidelity by the study nurse or Community Health Worker (CHW), data quality checks and results dashboard review.

The interim analysis will be for monitoring purposes from the Kobo toolbox dashboard. It will be used to determine acceptance and retention rates and forecasting trial period remaining and the proportion of the unreached or dropout rates. This being a behavioural intervention study, the researcher do not envisage a circumstance warranting prior termination of the trial due to internal factors.

## Analysis

The participant will form the unit of analysis. All analyses will be by intention to treat. Descriptive statistics will be tabulated for individual characteristics and summarized into frequencies and percentages for categorical variables and means, median, range and standard deviation for continuous variables.

Bivariate analysis of effects of sociocultural beliefs on PPFP uptake will be done and presented on two-by-two (2x2) tables with Chi-square being the inferential statistics. P-values will be used to assess the significance of homogeneity of proportions and odds ratio (OR) and 95% confidence interval (95% CI) will demonstrate the strength of the relationship. Binary logistic regression analysis will be done to adjust for confounders of sociocultural beliefs as determinants of PPFP uptake.

Fertility intentions for postpartum mothers will be measure by Likert scale [25] and disaggregated based on sociodemographic aspects and other individual characteristics. Intention to use PPFP will be simplified in proportions of the categorical *yes* or *no* to appointment for PPFP. Bivariate analysis of determinants of fertility intentions will be done and presented in two-by-two (2x2) tables with Chi-square being the inferential statistics. Binary logistic regression analysis will be done to adjust for confounders of determinants of fertility intentions.

Student t-test will be used to analyze perceived individual control of PPFP choice and level of fertility intention to ascertain the significance in differences in means for the determinants of perceived control of FP choice and level of fertility intention and significant determinants will be fitted in multilinear regression analysis to adjust for confounders.

Postpartum Family Planning uptake between control and intervention groups will be compared by use of simple clinical superiority in the proportions.

Bivariate analysis with Chi-square statistics will be used to analyse the determinants of Postpartum Family Planning uptake between control and intervention groups and this will form the selection criteria for inclusion in the final regression model.

## Ethics statement

The study has been approved by Masinde Muliro University of Science and Technology (MMUST) School of Graduate Studies (SGS). Ethical clearance has been obtained from the

MMUST Institutional Ethics Review Committee (IERC). An official data collection permission letter will be obtained from the County Director of Health and Sanitation. A research authorization and permit has been acquired from National Commission for Science, Technology and Innovation (NACOSTI). Signed written informed consent for participation will be obtained from all participants after they are introduced to the study and informed about their rights. To ensure confidentiality and privacy, the names of the participants will not be recorded in the CRFs and data collection will be in privacy. Principle of justice and impartiality will be adhered to by enabling equal opportunity of the target population to participate in the study by use of probability sampling.

## Discussion

### Protocol rigor

The dedicated study core team will comprise the PI and the 2 Co-PIs and they will be the custodians of study-specific procedures to assure the protocol and standard operating procedures are followed and data are accurately collected.

Standardized study-specific training, supervision, and oversight will be undertaken to ensure quality, consistency, and harmonized trial procedures and implementation.

The core team has the oversight of Trial title, sourcing for funding, protocol amendment, follow-up on data collection, management and recommendations by data management team, setting recruitment date, setting recruitment end date, monitoring actual recruitment rate versus the projected recruitment rate, consenting process, summarizing protocol deviations, site visits and site reporting of organizational problems or other trial issues.

Data management team will comprise the PI, Data Clerks at the sites, the study nurse and CHV. The team will set acceptance rate proportion (those who accepted to participate versus those sampled), monitoring retention (percentage of participants proceeding from recruitment, treatment to follow-up including missing outcome data), monthly forecast of recruitment for the trial period remaining, monitoring loss to follow-up as a proportion of the ones not yet reached by 16 weeks postpartum follow-up to those who were recruited, overseeing data management metrics i.e. rate of electronic data capture, return dates and rate of returns, number of completed follow-up, withdrawal rate, monitoring intervention fidelity by the study nurse or CHV, data quality checks and results dashboard review.

### Dissemination plans

After analysis of data results will be used by the core research team to develop manuscripts for publishing with peer reviewed journal and presentation in international conferences, targeting those involved in the FP service provision especially in low-resource settings as well as those who develop and advise on policies and guidelines in those settings. Equally, a full thesis report by the PI will be archived with MMUST thesis repository. There will be a feedback session with health care professionals, at operational and strategic levels, within the intervention region. The researchers will also disseminate findings in popular media portals like Twitter, Facebook and local dailies.

The protocols with be published in peer reviewed trials journal, but participant level data sets and analysis code can be shared upon reasonable written request.

### Limitations of the study design

Together with the measurement, estimator, assumption and strategy limitations of any study, limitations to this study and potential sources of bias include incomplete fidelity to study

guidelines by the implementers, potential loss to follow-up and the nationwide shortage of FP commodities. To minimize non-adherence to the protocols, training material will be supported with short videos and other text documents as reference for the study requirements. Equally site appraisals and client exit interviews will estimate protocol fidelity. Learning was standardized by an pre and posttest and those who only attained minimum set requirement for the test were allowed to proceed. To minimize loss to follow-up, mothers will be provided with clear follow-up instructions with an appointment sticker for the return date as well as called the afternoon before their final questionnaire visit. Equally, the visit was set on the 14th week postnatal which coincides with the third immunization visit (which is rated at 95% national compliance). A reasonable return travel stipend will be provided for the final questionnaire visit. As for the nationwide shortage of FP commodities, this applies evenly across the setting thus will not affect the net overall finding.

Hawthorne effect may be introduced especially in the control arm since some of the nurses and CHVs meet during dissemination of other health related material and may share experiences including current study. However, this might cut across board as CHVs will try to outdo the nurses in client follow up and vice versa transmitting the overall Hawthorne effect across board. This is not likely to significantly affect the difference in proportions of uptake.

The study's overall purpose is to find if the targeted FP information integration in antenatal care can have overall clinically significant effect on early postpartum uptake of FP. Since the

| Section/item (Item No) | Description |
|---|---|
| **Administrative information** | |
| Title (1) | Effectiveness of targeted antenatal family planning information provision on early postpartum family planning uptake in Kisumu County: Protocol for a Simple Randomized Control Trial (I DECIDE Study) |
| Trial registration (2a/b) | Pan African Clinical Trials Registry PACTR202109586388973 |
| | **Primary registry and trial identifying number:** Pan African Clinical Trial Registry PACTR202109586388973<br>**Date of registration in primary registry** 28 September, 2021<br>**Secondary identifying numbers/Acronym** I-decide<br>**Source(s) of monetary or material support:** The PI - Morris Senghor Shisanya (MSS)<br>**Primary sponsor:** The PI<br>**Secondary sponsor(s):** None<br>**Contact for public queries:** MSS, BScN, MScN [+254720640142] [senghormorris@gmail.com]<br>**Contact for scientific queries:** MSS, BScN, MScN [+254720640142] [senghormorris@gmail.com]<br>**Public title:** Effectiveness of targeted antenatal family planning provision on early postpartum family planning uptake in Kisumu County: Protocol for a Simple Randomized Control Trial (I DECIDE Study)<br>**Scientific title:** Targeted Antenatal Information Provision and Postpartum Fertility Decisions<br>**Countries of recruitment:** Kenya<br>**Health condition(s) or problem(s) studied:** Female Fertility, Contraceptives<br>**Intervention(s)**<br>**Active comparator:** Antenatal provision of Postpartum Family Planning (PPFP) Information and appointment setting for 14 weeks postpartum follow-up<br>**Control comparator:** Routine Antenatal care<br>**Key inclusion and exclusion criteria** Facilities: 4 primary health centres and 2 communities in the Kisumu East Sub-County. Health Centers must be: offering the continuum of ANC, delivery, and PNC; providing a selection of at least three modern contraceptive methods that are rated 2 or 1 on the Medical Eligibility Criteria (MEC) for Postpartum Contraceptives and referrals for other methods to clients; having adequate contraceptive commodity stock with no stock-outs of contraceptives during the preceding six months; having on average at least 10 deliveries per month; and willing to participate.<br>All pregnant women will be eligible to participate in the study if: They are in their second pregnancy trimester; The woman is attending ANC and has intention of attending PNC at the health centre; An informed consent is obtained; and Resides within 20km<br>Excluded from the study will be clients: in similar study; with latex sensitivity; not anticipating a male partner in the next 12 months; unable to complete consent form as determined by the study nurse or cha and whose only male partner has had a vasectomy.<br>**Study type** Interventional<br>**Allocation:** randomized<br>**Intervention model:** concurrent assignment<br>**Masking:** None<br>**Primary purpose:** prevention<br>Phase II<br>**Date of first enrolment** 20th February, 2022<br>**Target sample size** 246<br>**Recruitment status** Not Recruiting<br>**Primary outcome(s)** Nature of Fertility decisions postpartum. Primarily the uptake of immediate postpartum FP being the main outcome – 14 weeks postpartum<br>**Key secondary outcomes** Fertility intentions. Primarily whether they accept to book an appointment for postpartum family planning - 14 weeks postpartum<br>Knowledge of family planning methods – Any time after intervention |
| Protocol version (3) | 28th September 2021 |
| Funding (4) | The Trial will be funded by the principal investigator. |

**Fig 4. Trial registration summary.**

study targets those already attending ANC clinic, selection bias can influence generalization of the effectiveness of the approach to the general population of pregnancy mothers but not generalization to the health systems implementation of the same.

## Amendments to the study

The PI will be responsible for communicating important protocol modifications (eg, changes to eligibility criteria, outcomes, analyses) to relevant parties (e.g. study implementers, MMUST Institutional Ethics Review Committee (IERC), trial participants, Pan-African Clinical Trials Registry (PACTR), PLOS ONE, and NACOSTI).

## Termination

The end point adjudication team will comprise the PI, the facility in-charges of the study facilities, the sub-county RH coordinator.

## Trial status

This protocol is version number 1 PACTR202109586388973 of the 28[th] September 2021. The researchers started recruitment on 20[th] February, 2022 and expect closure by August 2022. Additional information about the trial registration is presented in Fig 4 and S1 Checklist.

## Supporting information

**S1 Video. Standardized tool for counselling.** This is a screen recording of the Kobo collect Standardized FP counselling guide to be used by the nurse or CHV during the counselling session in the intervention facilities.
(MP4)

**S1 Checklist. SPIRIT 2013 checklist: I decide project.** Recommended items to address in a clinical trial protocol and related documents.
(PDF)

**S1 Protocol.**
(DOC)

## Author Contributions

**Conceptualization:** Morris Senghor Shisanya, Mary Kipmerewo.

**Formal analysis:** Morris Senghor Shisanya, Collins Ouma.

**Investigation:** Morris Senghor Shisanya.

**Methodology:** Morris Senghor Shisanya, Collins Ouma.

**Project administration:** Morris Senghor Shisanya.

**Resources:** Morris Senghor Shisanya.

**Software:** Morris Senghor Shisanya.

**Supervision:** Collins Ouma, Mary Kipmerewo.

**Validation:** Morris Senghor Shisanya.

**Visualization:** Morris Senghor Shisanya.

**Writing – original draft:** Morris Senghor Shisanya.

**Writing – review & editing:** Morris Senghor Shisanya, Collins Ouma, Mary Kipmerewo.

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
