## [Decision Letter · Decision Letter 0]

18 Mar 2022

PONE-D-22-04234Effectiveness of targeted antenatal family planning information provision on early postpartum family planning uptake in Kisumu county: Study protocol for a Simple Randomized Control TrialPLOS ONE

Dear Dr. Shisanya,

Thank you for submitting your manuscript to PLOS ONE. After careful consideration, we feel that it has merit but does not fully meet PLOS ONE’s publication criteria as it currently stands. Therefore, we invite you to submit a revised version of the manuscript that addresses the points raised during the review process.

We look forward to receiving your revised manuscript.

Kind regards,

Walid Kamal Abdelbasset, Ph.D.

Academic Editor

PLOS ONE

Journal Requirements:

Reviewers' comments:

Reviewer's Responses to Questions

**Comments to the Author**

1. Does the manuscript provide a valid rationale for the proposed study, with clearly identified and justified research questions?

Reviewer #1: Yes

Reviewer #2: Yes

2. Is the protocol technically sound and planned in a manner that will lead to a meaningful outcome and allow testing the stated hypotheses?

Reviewer #1: Partly

Reviewer #2: Yes

3. Is the methodology feasible and described in sufficient detail to allow the work to be replicable?

Reviewer #1: Yes

Reviewer #2: Yes

4. Have the authors described where all data underlying the findings will be made available when the study is complete?

Reviewer #1: Yes

Reviewer #2: Yes

5. Is the manuscript presented in an intelligible fashion and written in standard English?

Reviewer #1: No

Reviewer #2: Yes

6. Review Comments to the Author

You may also provide optional suggestions and comments to authors that they might find helpful in planning their study.

Reviewer #1: *The authors have come with an interesting research topic that touches women’s health, paving a way to addressing the unmet contraceptive needs in women living in resource-limited regions. In general, the study protocol is well written, and it should be considered for further research undertaking provided that the suggested modifications from both the reviewer (s) and academic editor(s) are incorporated adequately. Below are the main comments of the review report.

*The researchers have declared that purpose of the study will be explained to the participants. I was wondering of a possibility of Hawthorne effect in which people behave differently (increased postpartum family planning uptake) once they know that they are being studied.

*The objective of the study (study aim) was added to the methods section of the protocol. Is that appropriate?

*How are the possibility of spillover effect and selective attrition will be addressed? How is the influence of external factors such as mass media and partner’s desire are considered to be handled?

*There are some punctuation errors. For instance, the word ‘respectively’ is not preceded with a comma throughout the document. The word ‘antenatal’ was capitalized inappropriately.

*Study aim section: the researcher or the researchers?

*List of abbreviations/acronyms should be revised. Abbreviations such as SGS, GATHER, BRAIDED should not be mentioned in the list as each one of them have been used only once and under brackets.

Reviewer #2: Dear PLOSE ONE team of editorials, thanks for the chance given to me to review a protocol manuscript titled “Effectiveness of targeted antenatal family planning information provision on early postpartum family planning uptake in Kisumu county: Study protocol for a Simple Randomized Control Trial”. The protocol will have its own importance for future better utilization of the early post partal family planning and maternal health in general. The following are my comments for further improvement of the protocol.

1. General Comments

• The problem statement should be very strong to ensure the need to study ANC FP information provision to improve early PPFP uptake.

• The problem statement should be in line with purpose statement.

• The justifications of the study are not justifications.

• Try to shorten and summarize the analysis section.

• The phases and its intended objective should be clear and concise.

• Are pregnant women best candidate for receiving FP information? How do you manage regret, depression or anxiety if they became pregnant by the failure of family planning?

• It should be ‘clustered randomized trial’.

2. Specific Comments

A. On The Methods Section

What is unique in standardized intervention for at ANC FP information provision to improve early PPFP uptake when compared with routine one?

Describe briefly the illegibility criteria i.e. If similar study is running in the health center?

What type of family planning? I guess it should be reversible long acing contraceptive to have public health significance and to lengthen the birth interval?

How do you analyze fertility intentions?

Is the sample adequate to ensure external validity?

What are the exclusion criteria’s?

What is the main purpose of the protocol development and what type of protocol is that? It is effectiveness. Hence, what are the pitfalls in detail for the low effectiveness of ANC FP provision to improve early uptake of PPFP?

Who are those information conveyers? What is their special quality?

What are the contents of the counselling tool [try to mention by table]?

What are the frequencies of conveying this advice?

When do we say early post-partum?

When do we say post-partal women have early PPFP uptake? Who measures that?

What extent of early PPFP uptake do you expect as a result of your intervention? Even did early PPFP uptake impact, outcome or an output?

How do you assess whether your project or other competing programs have impacted uptake of PPFP?

Did the feasibility will follow the protocol? If so what expected results will derive you to continue to the next phase?

Why don’t you use ecological framework? Have you consulted normalization theory? Is your comprehensive to cover all the stake holders of the post-partal family planning services actors?

Simply what is your unit of analysis?

What is your ratio of matching?

How did you assure data quality, mention in detail since it is plan?

Why you prefer to use t-test?

Generally, it is very good work with ample benefit that need brief refinement.

Regards,

7. PLOS authors have the option to publish the peer review history of their article (what does this mean?). If published, this will include your full peer review and any attached files.

Reviewer #1: **Yes: **Subah Abderehim Yesuf

Reviewer #2: No

---

## [Author Response · Author response to Decision Letter 0]

9 May 2022

This article being a study protocol does not report data and the data availability policy is not applicable to the article. However, reasonable request will be considered by the core study team after study closure and data management to provide the data that will be used in making inferences.

---

## [Decision Letter · Decision Letter 1]

23 May 2022

PONE-D-22-04234R1Effectiveness of targeted antenatal family planning information provision on early postpartum family planning uptake in Kisumu county: Protocol for a Simple Randomized Control Trial (I DECIDE Study)PLOS ONE

Dear Dr. Shisanya,

Thank you for submitting your manuscript to PLOS ONE. After careful consideration, we feel that it has merit but does not fully meet PLOS ONE’s publication criteria as it currently stands. Therefore, we invite you to submit a revised version of the manuscript that addresses the points raised during the review process.

We look forward to receiving your revised manuscript.

Kind regards,

Walid Kamal Abdelbasset, Ph.D.

Academic Editor

PLOS ONE

Journal Requirements:

Reviewers' comments:

Reviewer's Responses to Questions

**Comments to the Author**

1. Does the manuscript provide a valid rationale for the proposed study, with clearly identified and justified research questions?

Reviewer #1: Yes

Reviewer #2: Yes

2. Is the protocol technically sound and planned in a manner that will lead to a meaningful outcome and allow testing the stated hypotheses?

Reviewer #1: Yes

Reviewer #2: Yes

3. Is the methodology feasible and described in sufficient detail to allow the work to be replicable?

Reviewer #1: Yes

Reviewer #2: Yes

4. Have the authors described where all data underlying the findings will be made available when the study is complete?

Reviewer #1: Yes

Reviewer #2: No

5. Is the manuscript presented in an intelligible fashion and written in standard English?

Reviewer #1: Yes

Reviewer #2: Yes

6. Review Comments to the Author

You may also provide optional suggestions and comments to authors that they might find helpful in planning their study.

Reviewer #1: The authors have tried to tackle each comment raised by the previous review, and as a consequence, the protocol is now in good stand except for minor capitalization and spacing problems.

Reviewer #2: Dear PONE team of editorials, thank you for the chance given to me to review a manuscript titled “Effectiveness of targeted antenatal family planning information provision on early postpartum family planning uptake in Kisumu country: Protocol for a Simple Randomized Control Trial (I DECIDE Study)”. The paper will have contribution for designing appropriate and effective family planning information and further reached education after giving birth.

The following are my comments;

• Gaps related to the information need and the justification should be strong.

• What are the adverse consequences occurred due to the convey of the conventional [routine standard care] post-partum family planning information?

• The eligibility criteria are not very strong and binding.

• The tool needs face validation.

• The protocol should address;

i. Clear sampling without making ambiguity to the reader.

ii. What are the software’s that are used to calculate the sample size and population allocation.

iii. Face validity of the tool.

iv. Pre-test

v. The power and the effect expected due to the intervention/s

vi. Correction, revisions and actual effective implementations.

vii. Data collectors, their quality and data advanced data quality measures and its management.

viii. Dose, dosage, total duration of the intervention for both the case and the controls.

ix. Expected outcomes

Primary Outcomes

Secondary outcomes

x. Ethical Issues: Trial registration, ethics and its intervention.

Regards,

7. PLOS authors have the option to publish the peer review history of their article (what does this mean?). If published, this will include your full peer review and any attached files.

Reviewer #1: **Yes: **Subah Abderehim Yesuf

Reviewer #2: **Yes: **Ok

---

## [Author Response · Author response to Decision Letter 1]

4 Jun 2022

Thank you for the insights shared. Pleas find our responses below. Reviewer’s concerns are in bold and the responses are in normal font. 

The following are my comments;

• Gaps related to the information need and the justification should be strong – This has been noted, however, operational feasibility of integrating information giving during the antenatal period is the main intent of the study. Information as it were is already a proven means of increasing demand for FP but the main issue is how best should information be integrated in MCH services. Can deliberate antenatal integration be much better than routine way of doing it? 

• What are the adverse consequences occurred due to the convey of the conventional [routine standard care] post-partum family planning information? – Study’s aim to compare the two Standard/routine care vs ANC Integrated PPFP information giving. The reason for comparison is because routine care has resulted/posted low uptake of early PPFP. L79-86

• The eligibility criteria are not very strong and binding – Noted. This is a behavioural intervention and basically applying implementation science as opposed to biomedical science. This means that apart from exclusion criteria, we used WHO FP eligibility criteria for inclusion. 

• The tool needs face validation - Attached as S3 

• The protocol should address;

i. Clear sampling without making ambiguity to the reader – Done L165-166, &L230-232

ii. What are the software’s that are used to calculate the sample size and population allocation – based reference 21 L492/493 A simple excel based calculator was used to calculate the sample size based on reference 19 L487/488. 

iii. Face validity of the tool – See S2 attached

iv. Pre-test – Done

v. The power and the effect expected due to the intervention/s – Considered in formula as β (type II error set at 0.1 or 90% power) L181/182

vi. Correction, revisions and actual effective implementations – Were stipulated in the protocol registration with PACTR. So far no revisions have been made to the protocols registered with PACTR

vii. Data collectors, their quality and data advanced data quality measures and its management –Done L186-210, & L249-263

viii. Dose, dosage, total duration of the intervention for both the case and the controls – Done.L186-210

ix. Expected outcomes

Primary Outcomes – L197

Secondary outcomes – L205-2013

x. Ethical Issues: Trial registration, ethics and its intervention – L55 & L255-265

My Kind regards

---

## [Editor Report · Decision Letter 2]

13 Jul 2022

Effectiveness of targeted antenatal family planning information provision on early postpartum family planning uptake in Kisumu county: Protocol for a Simple Randomized Control Trial (I DECIDE Study)

PONE-D-22-04234R2

Dear Dr. Shisanya,

We’re pleased to inform you that your manuscript has been judged scientifically suitable for publication and will be formally accepted for publication once it meets all outstanding technical requirements.

Kind regards,

Walid Kamal Abdelbasset, Ph.D.

Academic Editor

PLOS ONE

Additional Editor Comments (optional):

Appreciating the authors for their responses to all comments. No further comments are required.
---

## [Editor Report · Acceptance letter]

4 Aug 2022

PONE-D-22-04234R2 

Effectiveness of targeted antenatal family planning information provision on early postpartum family planning uptake in Kisumu County: Protocol for a Simple Randomized Control Trial (I DECIDE Study) 

Dear Dr. Shisanya:

I'm pleased to inform you that your manuscript has been deemed suitable for publication in PLOS ONE. Congratulations! Your manuscript is now with our production department. 

Kind regards, 

on behalf of

Dr. Walid Kamal Abdelbasset 

Academic Editor

PLOS ONE